# Identification of Candidate Genes Regulating Carcass Depth and Hind Leg Circumference in Simmental Beef Cattle Using Illumina Bovine Beadchip and Next-Generation Sequencing Analyses

**DOI:** 10.3390/ani12091103

**Published:** 2022-04-25

**Authors:** Farhad Bordbar, Mohammadreza Mohammadabadi, Just Jensen, Lingyang Xu, Junya Li, Lupei Zhang

**Affiliations:** 1Key Laboratory of Animal Genetics Breeding and Reproduction, Ministry of Agriculture and Rural Affairs, Institute of Animal Sciences, Chinese Academy of Agricultural Sciences, Beijing 100193, China; farhadbordbar@agr.uk.ac.ir (F.B.); lijunya@caas.cn (J.L.); zhanglupei@caas.cn (L.Z.); 2Department of Animal Science, Faculty of Agriculture, Shahid Bahonar University of Kerman, Kerman 77951, Iran; mrm@uk.ac.ir; 3Center for Quantitative Genetics and Genomics, Aarhus University, 8210 Aarhus, Denmark; just.jensen@qgg.au.dk

**Keywords:** genome-wide association study, carcass depth, hind leg circumference, next-generation sequencing, Simmental beef cattle

## Abstract

**Simple Summary:**

The performance of genome-wide association studies with high-density single-nucleotide polymorphism panels and next-generation sequencing is a robust approach to the identification of genetic variants that can elucidate variation in complex diseases and economically important traits in farm animals. We identified many markers associated with crucial traits, such as muscle development in Simmental beef cattle. This information can be used to pinpoint candidate genes affecting traits of interest. Our results contribute to the clarification of molecular mechanisms underlying processes associated with hind-leg meat yield and carcass depth in beef cattle.

**Abstract:**

Genome-wide association studies are a robust means of identifying candidate genes that regulate economically important traits in farm animals. The aim of this study is to identify single-nucleotide polymorphisms (SNPs) and candidate genes potentially related to carcass depth and hind leg circumference in Simmental beef cattle. We performed Illumina Bovine HD Beadchip (~670 k SNPs) and next-generation sequencing (~12 million imputed SNPs) analyses of data from 1252 beef cattle, to which we applied a linear mixed model. Using a statistical threshold (*p* = 0.05/number of SNPs identified) and adopting a false discovery rate (FDR), we identified many putative SNPs on different bovine chromosomes. We identified 12 candidate genes potentially annotated with the markers identified, including *CDKAL1* and *E2F3*, related to myogenesis and skeletal muscle development. The identification of such genes in Simmental beef cattle will help breeders to understand and improve related traits, such as meat yield.

## 1. Introduction

Farm animals, particularly cattle, are of major economic importance in many parts of the world [1]. Beef cattle breeds are being developed genetically to maximize the efficiency of meat production [2] using breeding programs based on traits related to meat yield [3]. Recently, beef breeders have been inclined toward the augmentation of lean muscle mass growth to meet rapidly changing consumer demand for lean meat [4]. Thus, in the past several years, researchers have made more efforts to increase our understanding of genetic influences on economically important traits covering different aspects of meat production, especially muscle growth and development.

Simmental beef cattle, distributed in nearly all parts of the world, are bred primarily due to their high levels of growth and lean meat production under an appropriate diet [5,6,7]. Many studies have been conducted to examine the economically crucial traits of this breed, such as meat production [8], meat quality [9], carcass characteristics [10], and growth [11]. An understanding of the effects of genetic variation on meat quality, meat yield, and carcass quality will aid breeders’ implementation of methods to effectively address beef market demands.

The search for candidate genes via genetic association studies, conducted for almost all species [12], may be practical for the extension of our understanding of genes affecting economically important traits. Candidate genes may affect particular biological actions related to different aspects of a trait of interest, and their identification is an important step in efforts to identify quantitative trait loci (QTLs) that regulate the genetic variation of traits of interest [13].

The performance of genome-wide association studies (GWASs) using high-density single nucleotide polymorphism (SNP) panels is a robust approach to the identification of different parts of the genome and genetic variants that can elucidate variation in complex diseases and prominent or economically important traits in farm animals [14,15,16] and humans [17,18,19]. GWAS results can be sensitive to different factors, such as the sample size and the number of genes affecting traits of interest [20]. Next-generation sequencing (NGS) can be employed to access numerous genetic variants in close proximity to genes affecting traits of interest. Sharma et al. [21] used NGS to identify 18 mutations associated with Mendelian diseases in Hanwoo cattle; in addition, in the same cattle, 33 genes were found to be associated with domestication [22]. In Simmental beef cattle, NGS has been used to identify important genes that control hind leg width, such as *PLXNA4*, *ASB15*, *SLC13A1*, *NDUFA5*, *IQUB*, *LMOD2*, and *WASL* [23], and a promising candidate gene that controls net meat weight (*MTPN*) [24]. However, due to the high SNP density in NGS data, the calculation of marker effects is challenging [23,24,25].

Hindquarters from beef carcasses include valuable cuts, and their properties are thus important components of meat-related traits [23]. The hindquarter consists of connective tissues; the femur, tibia, and fibula; muscles including the semitendinosus, vastus lateralis, tensor fascia lata, gluteus medius, gastrocnemius, semimembranosus, and biceps femoris. Most of these muscles are skeletal and supply large amounts of meat. Muscle quantity has been reported to be closely related to the hind leg dimension [26]. In addition, the hind-leg muscle thickness correlated positively with the growth rate of European beef cattle [26].

In the past several years, genetic selection for carcass conformation traits has been practiced in most beef cattle populations. Carcass depth is an important carcass conformation trait that covers the principal muscle and bone tissues and thus is profitable in the beef industry. Han et al. [27] illustrated that it was associated with the *IGF2* gene, a key gene for muscle development and growth in farm animals [28]. However, the literature on carcass depth is limited, and more research is needed.

In this GWAS, we detected SNPs for the identification of candidate genes (via QTLs and markers) associated with hind leg circumference and carcass depth in Simmental beef cattle. Our results can contribute to the clarification of the molecular mechanisms underlying hind-leg meat yield and carcass depth.

## 2. Material and Methods

### 2.1. Ethics Statement

All procedures were conducted in accordance with the guidelines of China’s Council on Animal Care, and the study protocol was approved by the Institute of Animal Science of the Chinese Academy of Agricultural Science, Beijing, China (approval number: RNL09/07). Analyses associated with animals adhered to the rules of China’s Council on Animal Welfare.

### 2.2. Animal Resources and Phenotype Data

The Simmental beef cattle population examined in this study comprised 1346 cattle born between 2009 and 2015 in Ulgai, Xilingol League, Inner Mongolia, China. Following weaning, the cattle were transferred to the Jinweifuren Farm feedlot in Beijing. All calves were raised under standard conditions and fed a total mixed ration based on the nutritional requirements of beef cattle (NRC, 2018). At 16–18 months of age, the animals were slaughtered. Immediately after bloodletting, the target carcasses were refrigerated at 4°C until trait measurement (about 9 h after slaughter). The carcasses were hanged, and the carcass depth and hind leg circumference were measured in centimeters using a tape (Figure 1). Trait means, standard deviations, and ranges are shown in Table 1.

### 2.3. Genotype Examination and Quality Control

Illumina Bovine HD SNP Beadchip (770 k) and genotype examinations were performed using Illumina Genome Studio (Illumina Inc., San Diego, CA, USA). For quality control, individuals with high Pi-Hat values (representing sample duplication) were excluded, and animals and SNPs were excluded if the SNP call rate <90%, minor allele frequency (MAF) <5%, deviation from Hardy–Weinberg equilibrium detected (*p* < 10^−6^), or >10% genotype data missing. These criteria were assessed using PLINK software (ver. 1.07) [29]. After the exclusion of 94 cattle, 671,204 autosomal SNPs from 1252 cattle were analyzed.

### 2.4. Resequencing

Using genomic relationship data and Pi-Hat values, we chose 44 unrelated cattle for resequencing. The TIANamp Blood DNA kit (Tiangen Biotech Company Limited, Beijing, China) was used to extract genomic DNA. DNA concentrations and purity were quantified using the NanoPhotometer N50 device (Implen GmbH, Munich, Germany). DNA (1.5 μg) with an A260/280-nm absorbance ratio of 1.8–2 and A260/230-nm absorbance ratio of 2.0–2.2 was splintered using the Ultrasonicator S2 (Covaris, Woburn, MA, USA). Sequencing libraries were prepared using the Truseq Nano DNA HT sample preparation kit (Illumina Inc.) according to the manufacturer’s instructions of manufacturers. Each sample was given an index code for sequence recognition. The DNA samples were splintered into ≤350–base pair fragments by sonication, and DNA fragments were chosen for end polishing, addition of A-tailing, and sequencing with further PCR amplification using a full-length adapter. Sequencing libraries were prepared using the Illumina HiSeq 2500 system (Illumina Inc.). We obtained 9,621,765,847 reads and subjected them to quality control; low-quality reads with >10% unrecognized bases, >10% mismatches, or >50% poor-quality bases were removed. To facilitate down-stream analysis, SAM files were converted to BAM files. Then, we performed variant calling using samtools-0.1.19 mpileup [30]. Variants were excluded if an overall quality (QUAL) score of <20, a mapping quality (MQ) score of <30, and a read depth of <10. Moreover, we used the following proximity filters: if a variant was within 3 bp of another variant, the variant with the lower QUAL score was also excluded. Only bi-allele SNPs were retained for subsequent analyses. The average sequencing depth (representing the number of sequences for each base) of the samples was about 20×.

### 2.5. Variant Imputation

NSG-derived variants with MAFs > 0.05 (21,043,398 sequence variants from 44 animals) were subjected to imputation using BEAGLE (ver. 4.1) [31] with the default settings, based on algorithms determined by population data to infer genotypes for animals with missing information and haplotypes. We labeled genotypes for imputed sequence variants as 0 (homozygotes), 1 (heterozygotes), and 2 (alternative homozygotes). In total, 12,468,401 SNPs were obtained for chromosomes 1–29 from the RNA sequencing data with the use of the standard of imputation quality >0.1 [32].

### 2.6. Statistical Analysis

We employed a general mixed linear model based on the formula $y=1\mu +Xb+m_{j}b_{j}+Zu+e,$ where *y* is a vector phenotypic value and *μ* represents the population mean. We applied two fixed-effect variables for single marker regression; b reflects noise associated with fixed effects (weight, sex, fattening days, and birth year) and *bj* reflects SNP effects. *m_j_* is the vector for the *i*th marker and u reflects the polygenic effect assumed with *N (0*, *σ2K)*, where *K* corresponds to the kinship matrix and *σ^2^* is the additive genetic variance. Although all SNPs on autosomal chromosomes were qualified for inclusion in the analysis, SNPs on chromosomes with *m_j_* were excluded. *X* and *Z* are the incidence matrices connecting the phenotypic values to fixed and polygenic effects, respectively. *e* represents random residual effects assumed with$\;\;V\left(e\right)=I\sigma_{e}^{2}$, where *I* is an identity matrix and $\sigma_{e}^{2}$ is the residual variance.

Associated SNPs were identified using GenABEL (ver. 1.8-0) [33] in R, with the application of the significance criterion of Bonferroni-corrected *p* < 0.05 in accordance with division by the number of SNPs. For carcass depth, dependable QTLs and putative candidate genes were identified with a stringent significance threshold of 7.45 × 10^−8^. To obtain p values for the SNPs, t values were calculated. Considering the fact that Bonferroni correction outcomes are too stringent with low statistical power [34], we determined the threshold values for GWAS by adopting false discovery rate (FDR). The FDR was defined as 0.01, and the threshold p value was computed based on the formula P=FDR×n/m, where *n is* the number of *p* < 0.01, and *m* represents the total number of SNPs [35].

To detect candidate genes of interest, we used the UCSC genome browser (http://www.genome.ucsc.edu, Bos_taurus_UMD_3.1, accessed on 2 October 2020).

NGS dataset analysis is complex due to the enormous number of markers, which can make conventional p-value computation for SNPs difficult [23,24]. We coped with this problem by dividing chromosomes into the following segments: segment 1 consisted of chromosomes 1–10 with 5,830,727 SNPs, segment 2 consisted of chromosomes 11–20 with 4,063,690 SNPs, and segment 3 consisted of chromosomes 21–29 with 2,573,984 SNPs. Dependable QTLs and putative candidate genes were detected using the significance thresholds of *p* = 8.58 × 10^−9^ for segment 1, *p* = 1.24 × 10^−7^ for segment 2, and *p* = 1.95 × 10^−7^ for segment 3, based on the number of markers in each segment.

### 2.7. Heritability Estimation

The heritabilities for carcass depth and hind leg circumference were estimated using GCTA v1.93.1 software in accordance with Bovine HD SNP array and imputed NGS. The genomic heritability of each trait was calculated using $\;h^{2}=\sigma_{\alpha \;}^{2}/\;\sigma_{\alpha }^{2}+\sigma_{e}^{2}$.

## 3. Results

### 3.1. SNPs Associated with Carcass Depth

The estimated heritability for carcass depth was high (0.41 ± 0.06). With beadchip analysis, we identified several SNPs associated significantly with carcass depth (Table 2), including BovineHD0200025561 (*p* = 1.49 × 10^−14^) in candidate gene *CLK1* on chromosome 2 and BovineHD0200025553 (*p* = 3.4 × 10^−13^) and BovineHD0200025550 (*p* = 8.3 × 10^−13^) in candidate gene *BZW1*. Exploration of the region that is about 10 kb upstream of the marker BovineHD0200025562 (*p* = 1.49 × 10^−14^) led to the identification of candidate gene *PPIL3*. The marker BovineHD0500023139 (*p* = 2.17 × 10^−8^) was found in the gene *CCDC91* on chromosome 5. In addition, the candidate genes *DIP2B* and *LARP4* were identified about 3 kb upstream and 24 kb downstream of the marker BovineHD0500008673 (*p* = 1.3 × 10^−13^), respectively. On chromosome 17, the gene *c17h22orf31* was identified about 14 kb downstream of the SNP BovineHD1700020542 (*p* = 3.47 × 10^−8^). We also identified two significantly associated SNPs on chromosomes 10 (BovineHD1000031388; *p* = 1.17 × 10^−8^) and 26 (BovineHD2600013096; *p* = 5.58 × 10^−8^), but no associated candidate gene. Figure 2 shows the Manhattan plot for putative QTLs related to carcass depth.

### 3.2. SNPS Associated with Hind Leg Circumference

The estimated heritability for hind leg circumference was high (0.52 ± 0.12). Of the 9,621,765,847 raw reads acquired, 9,584,920,309 reads were examined after the application of quality control measures. In segment 1, we identified one SNP (rs91327124; *p* = 1.23 × 10^−9^) in BTA5 related to hind leg circumference, but no candidate gene in the vicinity of this SNP. In segment 2, several associated SNPs were identified (Table 3). The candidate gene *ZBTB16* was identified about 50 kb downstream of the most strongly associated SNP in this segment (rs25168454; *p* = 1.31 × 10^−10^), *NXPE4* was identified about 4.7 kb downstream of the SNP rs25439719 (*p* = 4.14 × 10^−8^), and the candidate gene *RBM7* was found about 20 kb upstream of the SNP rs25237349 (*p* = 1.12 × 10^−7^).

In segment 3, we found many putative QTLs on chromosomes 23, 26, and 29 (Table 4). Exploration around flanking SNPs in BTA26 and BTA29 revealed no related candidate genes. The only associated SNP in BTA23 (rs37207517; *p* = 8.61 × 10^−8^) was located in the gene *CDKAL**1*, which might be a crucial gene for Simmental beef cattle. About 47 kb upstream of this gene, the candidate gene *E2F3* was identified. Figure 3 shows the Manhattan plots for putative QTLs in segments 1–3 related to hind leg circumference.

## 4. Discussion

GWASs have been conducted for many animals (e.g., [36]) and plant (e.g., [37]) species, contributing to a better understanding of molecular mechanisms underlying economically important traits in farm animals, including beef and dairy cattle [15,23,24,38,39,40,41,42,43,44]. GWASs are a powerful approach to the detection of genome regions that carry causative genes [45,46,47]. We anticipate that NGS technology will aid scientists’ detection of currently unknown genetic variants [48]. In this study, NGS analysis yielded no results related to carcass depth, but the use of an SNP array did yield such results; the opposite was true for hind leg circumference.

Beadchip analysis revealed that the *CLK1* gene harbors the SNP BovineHD0200025561 in BTA2, associated most strongly with carcass depth. This gene codes for a protein kinase with binary specificity that phosphorylates tyrosine and serine/threonine residues [49]. Its N-terminal includes a regulatory portion containing receptors for *EGF* and other growth elements [50]. *EGF* plays a key role in cellular development by promoting cell proliferation and differentiation. *CLK1* has been reported to play a role in cancer due to the associations among *CLK1*, *ASF/SF2*, and *SRp20* [49] and in muscular dystrophy [51]. The *CLK1* inhibitor, *TG693,* is a splicing modulator with therapeutic potential for patients with Duchenne muscular dystrophy [51]. In pigs, high expression levels of the full-length splice form of *CLK1* were observed in the kidney, small intestine, and longissimus dorsi tissues [52]. In a rat model, *CLK* gene expression showed downregulation related to muscle fat during lactation [51]. In a gene ontology analysis, Lee et al. [53] detected a negative association between *CLK1* and the intermuscular fat content in Hanwoo cattle. In another study of Hanwoo cattle, *CLK1* was associated significantly with marbling (i.e., intramuscular fat) [54]. However, information on the effects of *CLK1* on meat quality and its relationship to cellular (especially muscle and bone) development in Simmental beef cattle is lacking. We hypothesize that *CLK1* plays a role related to meat quality and likely muscle cell development in Simmental beef cattle, but more research is needed to validate the relationship of this candidate gene to these traits.

In BTA2, we identified the candidate gene *BZW1*, which harbors two SNPs associated significantly with carcass depth (BovineHD0200025553 and BovineHD0200025550). This gene has been reported to be a proliferation regulator that promotes the progression of many aggressive cancers, such as salivary mucoepidermoid carcinoma [55]. *BZW1* plays an important role in cell cycle processes and transcriptionally manages the histone H4 gene in the G1/S phase. Chiou et al. [56] reported that its downregulation can repress lung adenocarcinoma metastasis. The roles of *BZW1* in tumor growth [57] and Alzheimer’s disease progression [58] have also been well demonstrated. However, very few studies have examined the function of this gene in farm animals. *BZW1* has been associated with bovine endometriosis [59], and Hoelker et al. [59] showed that its downregulation could influence the dynamic progression of embryos from cattle with subclinical endometritis. We hypothesize that *BZW1* is associated with carcass depth in Simmental beef cattle. More research is warranted to validate this association. We also identified the gene *PPIL3* about 10 kb upstream of the SNP BovineHD0200025562. *PPIL3* has been associated with estrogen receptor-negative breast cancer [60], but information on its function in farm animals is lacking. Thus, our identification of the potential association of *PPIL3* with carcass depth in Simmental cattle is novel. Further research is needed to corroborate this association.

In BTA5, we identified the candidate gene *CCDC91*, which harbors the SNP BovineHD0500023139, associated significantly with carcass depth. This gene is expressed in numerous cancer cell lines. *CCDC91* has been identified as a candidate gene associated with production traits (fat percentage) in dairy cattle [61], but few studies have examined this gene in farm animals. We hypothesize that this gene is associated with carcass depth in Simmental beef cattle. Moreover, we identified the candidate gene *DIP2B* about 3 kb downstream of the SNP BovineHD0500008673. In cattle, this gene has been associated with fatty acid content [62], an important trait affecting meat quality. *DIP2B* has also been associated with lipid metabolism, skeletal system development, and insulin signaling [62]. Information on its associations with traits in Simmental beef cattle is lacking; we hypothesize that it is related to carcass depth. We also identified the candidate gene *LARP4* about 24 kb upstream of the SNP BovineHD0500008673. This gene may be able to suppress cancer cell migration and invasion [63]. Egiz et al. [64] reported that *LARP4* repressed the motility and metastatic development of ovarian cancer cells. This gene has also been shown to control mRNA solidity and translation [65]. Zhou et al. [66] identified several candidate genes, including *LARP4B*, that influence milk protein composition traits in Holstein cattle. We hypothesize that *LARP4* is associated with carcass depth in Simmental beef cattle.

In the NGS analysis, we found that the SNP rs25168454 in BTA15 was associated most strongly with hind leg circumference. The candidate gene *ZBTB16* was located about 50 kb downstream of this SNP in segment 2. This conserved gene, broadly expressed in normal tissues such as cardiac and skeletal muscle [67,68] and also recognized as *PLZF* and *Zfp145*, acts as a crucial transcriptional suppressor or activator [69] that is required for various biological and developmental processes, including cellular apoptosis [70], hind limb construction [71], spermatogenesis [72], and anti-tumorigenesis activities [73]. It is also involved in processes such as metabolism regulation [74] and mesenchymal stem cell (MSC) differentiation [75], which is important during bone regeneration. In addition, *ZBTB16* controls homeostasis and the fate of hematopoietic stem cells [76]. Epigenomics and transcriptomics research has revealed its role in adipogenesis, as it controls fat accumulation [77,78], and it has been found to control thermogenics in brown fat and muscle [79]. Wei et al. [80] reported that the overexpression of *ZBTB16* enhances white adipogenesis and brown-like adipocyte formation in intramuscular preadipocytes in cattle. The bovine hind leg contains important muscles and bones, and its circumference is a good criterion for the assessment of traits related to bone and muscle development. Thus, we hypothesize that *ZBTB16* plays a role in bone and muscle development in Simmental beef cattle, but more research is needed to validate this hypothesis. In BTA15, about 4.7 kb downstream of SNP rs25439719, we found the candidate gene *NXPE4.* This gene has been recognized as a potentially promising prognostic biomarker for colorectal cancer [81]. Sarghale et al. [82] reported that *NXPE4* was a candidate gene located near SNPs related most closely to methane emission and fat traits in Holstein cattle. The candidate gene *RBM7* was also identified in BTA15, almost 20 kb upstream of the SNP rs25237349. The literature contains little information about this gene. A GWAS revealed that it was a candidate gene associated with milk production in Iranian buffalo [83]. We hypothesize that *NXPE4* and *RBM7* are associated with hind leg circumference in Simmental beef cattle.

In segment 3, we identified the very promising candidate gene *CDKAL1*, which harbors the only SNP associated with hind leg circumference in BTA23 (rs37207517). Several GWASs have revealed the role of *CDKAL1* in type 2 diabetes [84,85,86]. In addition, *CDKAL1* encodes the *CDK5* regulatory subunit–associated protein 1-like 1. This protein results from the translation of *CDKAL1* and is homologous to cyclin-dependent kinase 5 regulatory subunit–associated protein, a neuronal protein that can hamper *CDK5* activation [87]. *CDK5* can act as a myogenic protein kinase; it is required for myoblast differentiation and thus plays a key role in muscle myogenesis [88]. *CDKAL1* has also been associated with mitochondrial morphology and adipogenesis [89]. Thus, we hypothesize that it is associated with muscle development and hind leg circumference in Simmental beef cattle. We identified *E2F3*, another important candidate gene, about 47 kb upstream of *CDKAL1*. As a promoter of the muscle cell cycle, *E2F* is related to myogenesis [88]. Importantly, *E2F3* can induce myogenic differentiation [90]. Ma et al. [91] reported that microRNA-432 could hamper myogenesis by targeting *E2F3* in muscle cells. *E2F3* is associated with various other processes and plays an important role in the progression of a variety of cancers, such as bladder, breast, and prostate cancers [92,93,94]. Information about *E2F3* in Simmental beef cattle is lacking; we hypothesize that this gene is related to myogenesis in this breed.

## 5. Conclusions

Using the Illumina Bovine HD Beadchip and NGS dataset analyses, we identified many putative QTLs on several chromosomes that were associated with carcass traits in 1252 Simmental beef cattle. Several candidate genes related to carcass depth and hind leg circumference were identified. Among them, *CDKAL1* and *E2F3* may be associated with myogenesis and muscle development. Although the data were obtained by analyzing the genome of the Simmental breed, they can also be extended and validated in other cattle breeds. The identification of such genes helps breeders to understand and improve traits such as carcass depth and hind leg circumference.

## Figures and Tables

**Figure 1 animals-12-01103-f001:**
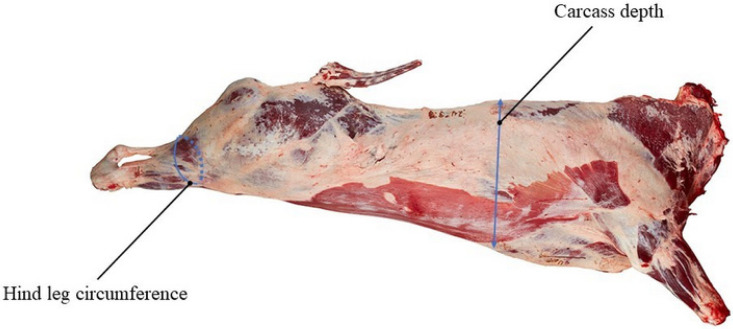
Measurement of hind leg circumference and carcass depth.

**Figure 2 animals-12-01103-f002:**
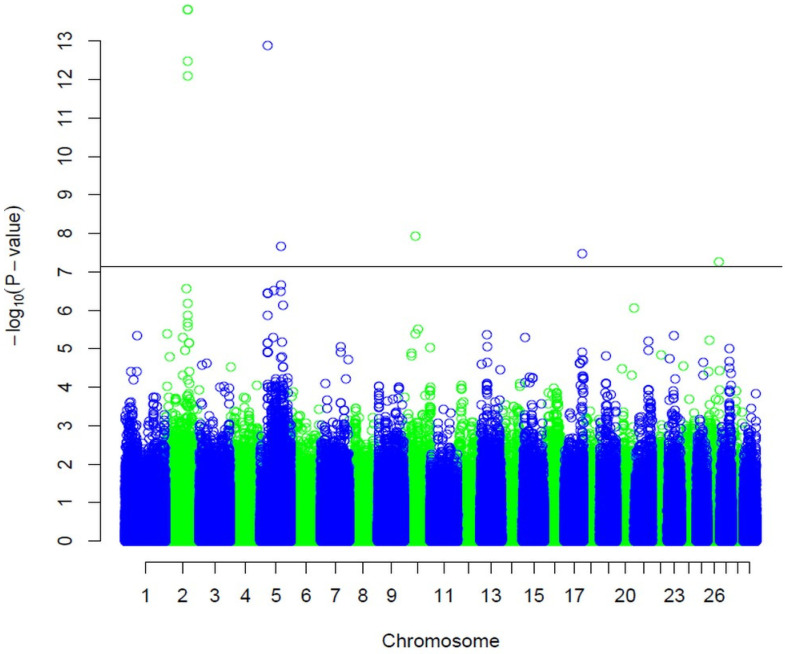
Manhattan plot of −log10 (p) for carcass depth. The horizontal line represents the threshold.

**Figure 3 animals-12-01103-f003:**
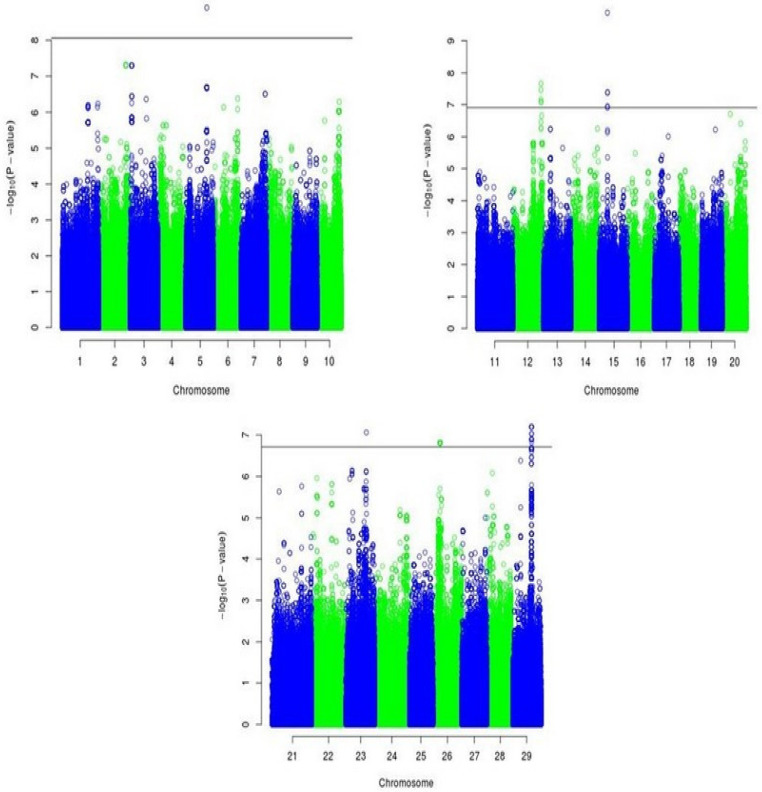
Manhattan plots of −log10 (p) for hind leg circumference in segments 1–3. The horizontal lines represent the thresholds.

**Table 1 animals-12-01103-t001:** Mean carcass depth and hind leg circumference.

Trait	Mean (cm)	Standard Deviation (cm)	Maximum (cm)	Minimum (cm)
Carcass depth Circumference of hind leg	63.88 68.01	4.62 16.58	82 110	13 22

**Table 2 animals-12-01103-t002:** SNPs associated with carcass depth, determined by GWAS analysis.

SNP	Chromosome	Position	*p*-Value
BovineHD0200025561 BovineHD0200025562 BovineHD0200025553 BovineHD0200025550 BovineHD0500008673 BovineHD0500023139 BovineHD1000031388 BovineHD1700020542 BovineHD2600013096	2 2 2 2 5 5 10 17 26	89923045 89926675 89896059 89887085 29582709 81891942 44213755 70557785 46267236	1.49 × 10^−14^ 1.49 × 10^−14^ 3.4 × 10^−13^ 8.3 × 10^−13^ 1.3 × 10^−13^ 2.17 × 10^−8^ 1.17 × 10^−8^ 3.47 × 10^−8^ 5.58 × 10^−8^

**Table 3 animals-12-01103-t003:** SNPs in segment 2 associated with hind leg circumference, as determined by NGS dataset analysis.

SNP	Chromosome	Position	*p*-Value
rs84436800 rs84450386 rs84441028 rs84429941 rs84428499 rs25168454 rs25439719 rs25457101 rs25237349 rs25224780	12 12 12 12 12 15 15 15 15 15	84436800 84450386 84441028 84429941 84428499 25168454 25439719 25457101 25237349 25224780	2.17 × 10^−8^ 3.45 × 10^−8^ 6.93 × 10^−8^ 8.13 × 10^−8^ 8.24 × 10^−8^ 1.31 × 10^−10^ 4.14 × 10^−8^ 4.14 × 10^−8^ 1.12 × 10^−7^ 1.22 × 10^−7^

**Table 4 animals-12-01103-t004:** SNPs in segment 3 associated with hind leg circumference, as determined by NGS dataset analysis.

SNP	Chromosome	Position	*p*-Value
rs37207517 rs12211246 rs12211363 rs12211411 rs34314488 rs34314780 rs34329034 rs34330397 rs34417763 rs34423556 rs34376791 rs34381008 rs34381454 rs34381516 rs34416372	23 26 26 26 29 29 29 29 29 29 29 29 29 29 29	37207517 12211246 12211363 12211411 34314488 34314780 34329034 34330397 34417763 34423556 34376791 34381008 34381454 34381516 34416372	8.61 × 10^−8^ 1.54 × 10^−7^ 1.54 × 10^−7^ 1.54 × 10^−7^ 6.32 × 10^−8^ 6.32 × 10^−8^ 6.32 × 10^−8^ 6.32 × 10^−8^ 9.3 × 10^−8^ 9.3 × 10^−8^ 1.27 × 10^−7^ 1.27 × 10^−7^ 1.27 × 10^−7^ 1.27 × 10^−7^ 1.5 × 10^−7^

## Data Availability

The data underlying this study have been uploaded to Dryad. The raw genotype data are accessible using the following doi: 10.5061/dryad.4qc06.

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
