# Peer review of "Identification of Candidate Genes Regulating Carcass Depth and Hind Leg Circumference in Simmental Beef Cattle Using Illumina Bovine Beadchip and Next-Generation Sequencing Analyses"

_animals, 2022, doi:10.3390/ani12091103_

Round 1

Reviewer 1 Report

In the paper, in one hand the authors used Illumina Bovine HD SNP Beadchips (770 k) to analyze 1252 cattle genomes. After analyses they identified several SNPs associated significantly with carcass depth but neither associated with hind leg circumference. Close to theses SNPs the authors found 6 genes putatively associated with carcass depth (CLK1, BZW1, PPIL3, CCDC91, DIP2B, LARP4)
On the other hand, based in NGS analysis, they found 26 SNPs associated and 5 genes putatively associated with hind leg circumference (ZBTB16, NXPE4, RBM7, CDKAL1, E2F3)
In the discussion, the authors speculate on the functions of these genes with an extensive analysis in different species, but in spite of this, they conclude that more studies are necessary to verify that these genes have relevant implications in these economically traits for bovine.
Although the data were obtained by analyzing genomes of the Simmental breed,  I think that the discussion and conclusions should not be restricted to this breed and should be extended to all cattle breeds.

Author Response

Reviewer: 1

We would like to extend our deepest gratitude to you, due to your great comment and suggestion. Undoubtedly, your suggestions and requests put this paper in higher level and more worthy for publication. Please see our answers to your suggestion below:

Your comment: Although the data were obtained by analyzing genomes of the Simmental breed, I think that the discussion and conclusions should not be restricted to this breed and should be extended to all cattle breeds

Response: Thank you so much. We designated one part addressing your point in conclusion as follows: Although the data were obtained by analyzing genome of the Simmental breed, they can also be extended and validated in other cattle breeds. Please see the line 327.

Reviewer 2 Report

Bordbar et al. performed the GWAS for two traits using the HD and NGS data. The manuscript describes a technically sound piece of scientific research with data that supports the conclusions. Experiments have been conducted rigorously with enough sample sizes for GWAS. The conclusions are drawn appropriately based on the data presented. However, it is not clear about genotypes and numbers of animals for each trait. Below, I have some minor comment

The authors should specify the traits in the title. It is very vague to name “economically important traits” in the title. Even I am not sure if these traits are in the selection indexes.

Line 24-25: it is not clear about the method. The authors performed GWAS using both types of data and using the imputed data.

Line 26: Specify the method for stringent threshold. In deep, I did not agree it is stringent enough as the authors used the chromosome segments for GWAS.

Line 28: “associated”, the author might change to “annotated”

Keywords: Might add one or two keywords for the traits and remove some such as SNPs

Line 68-69: The authors might update information about the numbers of QTL or QTN using the animal genome QTL database.

Line 81: The authors should be consistent in using the gene names or gene symbols (insulin-like growth

factor 2 gene). Also, the gene name should not be in Italics.

Line 92: Add the approval number if it exists.

Line 108-109: The sentence is not complete and not clear.

Line 109-113: might remove “was” in some section.

Line 128-130: It is completely lacking the variant calling sections. How did the authors reach this number of 21 Mil SNPs in line 135?

Line 135: Change sequence to sequence variants or NSG derived variants.

Line 150: What did the authors mean by “SNPs on chromosomes with mj were excluded”

Line 155: How many SNPs did authors use for calculating Bonferroni-corrected p < 0.05

Line 159: Add the version and date for UCSC Brower.

Line 160-165: I did not understand why the authors used the segments only for “hind leg circumference” not for carcass depth.

Figure 3: The authors could combine the results in one Manhattan plot. It is not necessary to plot in three segments.

Line 238 and others: SNP names do not need to be in Italics.

Line 256-257: Ref 59 is not necessary; I do not think the trait has some link to cancer study.

Author Response

Reviewer: 2

We would like to extend our deepest gratitude to you due to your great comments and suggestions. Undoubtedly, your suggestions and requests put this paper in higher level and more worthy for publication. Please see our answers to your suggestions and requests. We tried our best to fulfill your requirements and hope we have done this job as you wish.

Your comment: The authors should specify the traits in the title. It is very vague to name “economically important traits” in the title. 

Response: Thank you so much. It was revised and the name of traits were mentioned. Please see the line 2-4.

Your comment: Line 24-25: it is not clear about the method. The authors performed GWAS using both types of data and using the imputed data.

Response: Thank you so much. True. Let me explain here a little bit. First, we performed Illumina Bovine HD Beadchip (~670 k SNPs) to get access to important markers and candidate genes. Subsequently, we used NGS dataset to get more variants in related traits.

Your comment: Line 26: Specify the method for stringent threshold. In deep, I did not agree it is stringent enough as the authors used the chromosome segments for GWAS.

 Response: Thank you so much. It was revised please see the line 26, 27.

Your comment: Line 28: “associated”, the author might change to “annotated”

Response: Thank you so much, it was revised. Please see the line 28.

Your comment: Keywords: Might add one or two keywords for the traits and remove some such as SNPs

Response: Thank you so much. It was revised. Please see the line 32, 33.

Your comment: Line 68-69: The authors might update information about the numbers of QTL or QTN using the animal genome QTL database.

Response: Thank you so much for your great comment, absolutely. However, the point we wanted to address here was about the challenges using NGS dataset with high SNP density (around 12 million) which was really challenging and it was impossible to continue even with Linux. Therefore, we decided to divide all these SNPs into 3 segments and then analyze marker effects.

Your comment: The authors should be consistent in using the gene names or gene symbols (insulin-like growth factor 2 gene). Also, the gene name should not be in Italics

Response: Thank you so much. Sure. It was changed to gene symbol and its format become italic. Please see the line 82.

Your comment: Line 92: Add the approval number if it exists.

Response: Thank you so much. After talking to our group, I noticed we do not have such approval numbers and we submit our related papers with such information without approval code.

Your comment: Line 108-109: The sentence is not complete and not clear.

Response: Thank you so much. Actually, the sentence we brought indicates Illumina Bovine HD SNP Beadchip (770k) and genotype analyses were performed in Illumina Genome Studio. In addition, to address your point completely, we revise the sentence making it better for understanding. Please see the line 109-110.

Your comment: Line 109-113: might remove “was” in some section.

Response: Thank you so much. “Was” removed. Please see the line 112-114.

Your comment: How did the authors reach this number of 21 Mil SNPs in line 135?

Response: Thank you so much. It was revised and one more reference added. Please see the line 129-137.

Your comment: Line 135: Change sequence to sequence variants or NSG derived variants.

Response: Thank you so much. It was revised. Please see the line 141.

Your comment: Line 150: What did the authors mean by “SNPs on chromosomes with mj were excluded”

Response: Thank you so much for your great question. In the model we use single marker regression and correct for population structure using the u random effect. The random effect u have a covariance matrix based on SNP sets that do not include the chromosome being investigated. The reason for this is to avoid including very many SNPs in close LD with the SNP under investigation but at the same time take account of population structure and long range LD across chromosomes

Your comment: Line 155: How many SNPs did authors use for calculating Bonferroni-corrected p < 0.05

Response: Thank you so much. As explained, the  number of SNPs for segment 1 consisted of 5,830,727 SNPs, segment 2 consisted of 4,063,690 SNPs, and segment 3 consisted of 2,573,984 SNPs. These SNPs were used to calculate Bonferroni correction.

Line 159: Add the version and date for UCSC Brower.

Response: Thank you so much. It was revised. Please see the line 165.

Your comments: Line 160-165: I did not understand why the authors used the segments only for “hind leg circumference” not for carcass depth.

Response: Thank you so much for your great question. We used this grouping for both  traits actually. To address your point completely, we removed sentence “For the hind leg circumference”. Please see the line 171.

Your comment: Line 238 and others: SNP names do not need to be in Italics.

Reponse: Thank you so much. It was all revised throughout the paper.

Your comment: Line 256-257: Ref 59 is not necessary; I do not think the trait has some link to cancer study.

Reponse: Thank you so much. It was removed. Please see the line 263.

Reviewer 3 Report

The aim of this study was to identify single nucleotide polymorphisms (SNPs) and candidate genes potentially related to carcass depth and hind leg circumference in Simmental beef cattle. The author performed Illumina Bovine HD Beadchip (~670 24 k SNPs) and next-generation sequencing (~12 million imputed SNPs) analyses of data from 1,252 25 beef cattle.

The author identified 12 candidate genes potentially associated with the markers identified, including CDKAL1 and E2F3, related to myogenesis and skeletal muscle development.

These genes together with those published in reference 23 and 24 by the same authors are useful information for the breeder.

The work is extensive and provides many data. The authors have investigated the identified genes thoroughly, so that their hypotheses that these genes are associated with hind leg circumference or carcase depth are plausible.  

I accept the Manuscript in the present form.

Author Response

Thank you so much and we are very privileged that you accept this manuscript.